# Topologic Efficiency Abnormalities of the Connectome in Asymptomatic Patients with Leukoaraiosis

**DOI:** 10.3390/brainsci12060784

**Published:** 2022-06-15

**Authors:** Shun Yao, Hong-Ying Zhang, Ren Wang, Ding-Sheng Cheng, Jing Ye

**Affiliations:** 1Department of Medical Imaging, Clinic Medical School, Yangzhou University, Northern Jiangsu Province Hospital, Yangzhou 225000, China; yshhnl@163.com (S.Y.); zhying11@aliyun.com (H.-Y.Z.); 2Department of Neurology, Clinic Medical School, Yangzhou University, Northern Jiangsu Province Hospital, Yangzhou 225000, China; swiget@126.com; 3Department of Medical Engineering, Clinic Medical School, Yangzhou University, Northern Jiangsu Province Hospital, Yangzhou 225000, China; willbern44@126.com

**Keywords:** leukoaraiosis, diffusion kurtosis imaging, brain networks, graph theory

## Abstract

Leukoaraiosis (LA) is commonly found in aging healthy people but its pathophysiological mechanism is not entirely known. Furthermore, there is still a lack of effective pathological biomarkers that can be used to identify the early stage of LA. Our aim was to investigate the white matter structural network in asymptomatic patients with the early stage of LA. Tractography data of 35 asymptomatic patients and 20 matched healthy controls (HCs) based on diffusion kurtosis imaging (DKI) were analysed by using graph theory approaches and tract-based spatial statistics (TBSS). Diffusion parameters measured within the ALAs and HCs were compared. Decreased clustering coefficient and local efficiency values of the overall topological white matter network were observed in the ALAs compared with those of the HCs. Participants in the asymptomatic group also had lower nodal efficiency in the left triangular part of the inferior frontal gyrus, left parahippocampal gyrus, right calcarine fissure and surrounding cortex, right temporal pole of the superior temporal gyrus and left middle temporal gyrus compared to the ALAs. Moreover, similar hub distributions were found within participants in the two groups. In this study, our data demonstrated a topologic efficiency abnormalities of the structural network in asymptomatic patients with leukoaraiosis. The structural connectome provides potential connectome-based measures that may be helpful for detecting leukoaraiosis before clinical symptoms evolve.

## 1. Introduction

Leukoaraiosis (LA), which is defined as white matter abnormalities on CT or MR brain scans, is common in the aging population [1]. LA has been increasingly recognized to be attributed to cognitive dysfunction or dementia, stroke and some neurodegenerative disorders [2,3,4]. Given the potential aggressiveness of LA, a growing number of studies were performed to identify effective pathological biomarkers that can be diagnosed in symptomatic patients with LA [5,6]. However, few studies focused on the disruption of the brain network in the early stage of LA. Because structural changes are often accompanied by cognitive impairment and motor dysfunction in white matter (WM) lesion patients, investigating asymptomatic patients with early-stage leukoaraiosis may be beneficial for obtaining a better understanding of the pathogenesis of LA [7].

Recently, as scanning and analysis measures of MRI have developed rapidly, some advanced methods have emerged, such as diffusion tensor imaging (DTI). Structural disruptions in the relationship among anatomically distinct brain regions occur in patients with cerebral small-vessel disease from the data of diffusion tensor imaging, supporting the idea that LA might be attributable to a state of altered brain connectivity [8]. White matter structural connectome can be analysedby applications of diffusion magnetic resonance imaging post-processing and could be a clinically applicable measure for early LA diagnosis.

Although DTI has been very sensitive and popular for characterizing tissue microstructure since it was introduced, the limitation of DTI cannot be ignored due to its inability to account for the non-Gaussian diffusion presence of barriers [9]. Diffusion kurtosis imaging (DKI), which is composed of diffusion and kurtosis measurements, is proposed as an advanced diffusion MRI technique for white matter fiber tractography. It could characterize the non-Gaussian diffusion effects by including multiple b values and provide more accurate and richer diffusion information about the microstructure [10,11]. Thus, DKI was used to explore many neurodegenerative disorders [12].

In this study, we focused on asymptomatic subjects with leukoaraiosis (ALAs) accompanying a Fazekas grade 1 to 2 for white matter hyperintensity (WMH), which is defined as showing neuroimaging evidence of LA prior to the development of any overt clinical symptoms [13], to investigate the alterations in microstructural changes of major WM tracts and the disruption of the brain topologic network using DKI data.

## 2. Materials and Methods

### 2.1. Patients

Thirty-five ALAs and twenty healthy controls (HCs) were consecutively enrolled in the study from another study about neurodegenerative disorders after undergoing comprehensive physical, neurological and neuropsychological tests (Mini-Mental State Examination (MMSE) and Montreal Cognitive Assessment (MoCA)). The exclusion criteria included WM lesions with a history of neurodegenerative diseases, such as multiple sclerosis and Alzheimer’s disease. The exclusion criteria were metabolic disorder and systemic inflammatory disease, as well as other psychiatric comorbidities or severe cognitive impairment.

### 2.2. MRI Acquisition

All subjects were imaged with a 3.0 T MRI scanner (Discovery MR750, GE Healthcare) with an 8-channel head coil. The imaging protocol included 3D T1-weighted imaging (3D-T1WI), T2-weighted imaging (T2-WI), T2-fluid-attenuated inversion recovery (FLAIR) and a DKI sequence.

The 3D-T1WI sequence was obtained using an Magnetization Prepared Rapid Gradient Echo (MPRAGE) sequence in the sagittal plane with the following parameters: repetition time (TR)/echo time (TE) = 8.07/3.68 ms, flip angle = 6°, acquisition matrix = 256 × 256, FOV = 256 mm, slice thickness = 1 mm and resultant voxel size = 1 × 1 × 1 mm ^3^. Axial T2-WI was obtained using a fast spin-echo (FSE) sequence (TR/TE = 6500/102 ms, FOV = 240 × 240 mm ^2^, 5 mm thick slices). An axial T2-FLAIR sequence with parameters TR/TE = 9000/87 ms, FOV = 240 × 240 mm ^2^ and 5 mm thick slices. DKI data were obtained using a spin-echo single-shot echo planar imaging (EPI) sequence (TR/TE = 5800/77 ms; FOV = 256 × 256 mm; matrix = 128 × 128; slice thickness, 3 mm without gap; 30 encoding diffusion directions with two diffusion weightings (b = 1250, 2500 s/mm ^2^) for each direction and three b = 0 s/mm ^2^ volumes).

The location and size of the WMH on T2-FLAIR were evaluated in all subjects by two radiologists who each had more than 5 years of experience. Given that the volume of each segmented lesion was very small, we simply used the number of WMHs as the WMH size. The locations of WMHs were presented in the periventricular, paraventricular, frontal lobe, parietal lobe and temporal lobe. The number of WM lesions was not correlated with the age across the patients.

### 2.3. DKI Analysis

All diffusion datasets started with eddy current distortions and head motion artifact correction. Then, DKI and DTI parameters were processed using the Diffusion Kurtosis Estimator (DKE) (http://www.nitrc.org/projects/dke, accessed on 15 May 2022). Tensor of diffusioin (DT) and tensor of kurtosis (KT) parameters were extracted in DKE using the CLLS-QP algorithm. The DT and KT parameterized the DKI model and were used to build WM connectivity networks [14]. These parametric maps included DTI metrics of mean diffusivity (MD), axial diffusivity (AD), radial diffusivity (RD), fractional anisotropy (FA) and the corresponding DKI metrics of mean kurtosis (MK), axial kurtosis (AK), radial kurtosis (RK) and kurtosis fractional anisotropy (KFA) [14].

Each high-resolution 3D-T1 weighted image was first co-registered into the diffusion space in reference to non-b value images using FLIRT [15]. Then, a nonlinear transformation algorithm was used to transform the co-registered T1-weighted images to the Montreal Institute of Neurology (MNI) space by T1 template in MNI. [16]. The corresponding transformation matrices were then used to warp the Automated Anatomic Labeling (AAL) template from the standard MNI space and move it into the individual’s native DKI space.

To construct the DKI WM network, the AAL90 template was used to define the brain nodes. The main difusion direction of each voxel in the diffusion images were computed by a deterministic tractography algorithm seeding each voxel with a fractional anisotropy value > 0.2 and an angle > 45°. The weighting of each edge was defined as the number of fibers connecting a pair of brain nodes. The weighting of the edges were calculated as fiber numbers (FNs) and mean diffusion measures, including FA, MD, AD, RD, MK, RK, AK and KFA. Then all weighted 90 × 90 white matter connectome networks were acquired using PANDA (https://www.nitrc.org/projects/panda, accessed on 15 May 2022) [17]. The results of each participants were visually examined by an experienced neuroscientist. Then, the network indices were estimated by using Gretna (http://www.nitrc.org/projects/gretna/, accessed on 15 May 2022) and performed visually using BrainNet Viewer (http://www.nitrc.org/projects/bnv/, accessed on 15 May 2022) [18,19].

### 2.4. Topological Parameters

The clustering coefficient (Cp) and characteristic path length (Lp) were initially proposed by Watts and Strogatz [20], which may be the two key metrics for describing the small-world network [21]. Cp indicates the extent of local interconnectivity or cliquishness in a network. As for Lp, it quantifies the ability for information propagation in parallel in a network. Furthermore, a simple quantitative metric, small-worldness (σ), can be summarized from Cp and Lp measurements (σ = Cp/Lp), which is generally > 1 for small-world networks [21]. A previous study revealed that the global efficiency (Eglob) represents the global efficiency of transferring the parallel information in the network, and the local efficiency (Eloc) is associated with the fault tolerance of the network. It represents the efficiency of the communication between the first neighbors of i if i is removed [21]. The nodal efficiency (nodal (i)) measures the average shortest path length between a given node i and all of the other nodes in the network. In graph theory, betweenness centrality is a measure of centrality in a graph based on the shortest paths. It represents the degree of the nodes that stand between each other and quantifies the importance of a vertex for the information flow in a network [22]. In present study, all the parameters were estimated, also taking into account the weighted coefficient, in a wide range of sparsity threshold (5–40%).

### 2.5. TBSS

The WM status was assessed using tract-based spatial statistics (TBSS) for DKI and DTI topographies. Voxelwise statistical analysis of the FA data was carried out using TBSS (part of FSL (FMRIB Software Library)). All FA images from subjects were registered to the FMRIB58 FA template in the MNI space using a non-linear image registration toolkit. A mean FA image was calculated and used to create a mean WM FA skeleton. Then, a permutation-based interference tool and the Randomise tool were used to compare the FA data for skeletonized voxelwise statistical analysis. A restrictive statistical threshold was used (threshold-free cluster enhancement threshold, *p* < 0.05, corrected for multiple comparisons). All other parametric maps were then analyzed by repeating the same steps above separately.

### 2.6. Statistical Analysis

All continuous variables were tested for a normal distribution within groups using the Kolmogorov–Smirnov test. Demographic data, including age and years of education, were compared between ALAs and HCs by using a two-sample *t*-test. Sex and hypertension distribution were compared by using the chi-square test. An analysis of covariance (ANCOVA) was performed to compare the between-group differences in the small-world parameters and network efficiency of the WM structural networks on each diffusion metric. Age and sex were taken as covariates in this model. In addition, a false-discovery rate (FDR) method was performed to control for the error of multiple comparisons in both global and nodal network measurements. The threshold set to ≤0.05 was considered statistically significant (more details of the FDR method could be found in (https://github.com/sandywang/GRETNA, accessed on 15 May 2022).

### 2.7. Reproducibility Analysis

To evaluate the effects of different thresholds on the network analysis, we performed repeated network analyses with different thresholds by setting the threshold values of the number of fiber bundles to range from 1 to 5. We found that the threshold procedure did not significantly affect our results.

## 3. Results

### 3.1. Demographic Data

The demographic data of the ALAs and the controls are listed in Table 1. There were no significant differences with respect to age, gender, hypertension and years of education in the ALAs compared with the healthy controls (all *p* > 0.05). In addition, the clinical profiles including the scores of the MMSE and MoCA showed no significant differences between the two groups (all *p* > 0.05).

### 3.2. Overall Topology of WM Networks

Both the ALAs and HCs showed small-world network properties in their WM connectivity networks constructed by using determined tractography algorithm (σ > 1). ALAs showed significantly decreased Cp values compared with the controls as shown in the MK, AK and RK metrics belonging to DKI (*p* < 0.05, FDR-corrected). The Lp values were not significantly different in any of the DKI metrics. Eloc and Eg were computed to represent the efficiency of WM networks. As shown in Table 2, no significant alterations in Eg were observed in the ALAs (*p* > 0.05) whilethe ALAs showed decreased Eloc values compared with the outcomes of participants in the control group in the DKI metrics including MK, AK, RK and KFA metrics (*p* < 0.05, FDR-corrected).

### 3.3. Alterations in Nodal Efficiency

In the structural connectivity network, we identified brain regions showing significant between-group differences in at least one nodal metric. In addition, the correlation matrix of DKI and DTI showed different patterns, such as MK/MD metrics (Figure 1). Compared with the HCs, decreased nodal efficiency of the ALAs was observed in the left triangular part of the inferior frontal gyrus (IFGtriang.L, *p* = 0.002, FDR-uncorrected), left parahippocampal gyrus (PHG.L, *p =* 0.006, FDR-uncorrected), right calcarine fissure and surrounding cortex (CAL.R, *p =* 0.006, FDR-uncorrected), right temporal pole of the superior temporal gyrus (TPOsup.R, *p* = 0.005, FDR-uncorrected) and left middle temporal gyrus (MTG.L, *p =* 0.045, FDR-uncorrected) (Figure 2). Although no results showed significance after the FDR correction, the different trends observed in the ALA and HC groups also provided evidence that advanced our understanding of the pathophysiological mechanism of asymptomatic LA. No significant differences in betweenness centrality, degree centrality or characteristic path length were observed between participants in the ALA and control groups.

### 3.4. Hub

In the present study, the hub regions were determined by sorting nodal betweenness centrality values of MK metric networks. We identified the hub regions as having a nodal betweenness at least 1.5 times greater than the average betweenness values of the network across all regions [23].

Participants in both ALA group and healthy control group revealed similar hub distributions by computing the network of each group, distributions mainly located in bilateral superior frontal gyrus, anterior nucleus, putamen, left insular lobe, middle occipital gyrus, infratemporal gyrus and right hippocampus (Figure 3). Interestingly, several hubs, including the right insula, superior parietal gyrus and left middle temporal gyrus, but were only found in the healthy control group. Moreover, hubs in the right postcentral gyrus and the left hippocampus were only shown in participants in the ALA group.

### 3.5. WM Abnormalities Identified by TBSS

TBSS analysis revealed no differences (i.e., MD and MK) in the ALAs compared with that in the HCs.

## 4. Discussion

In our study, we used the DKI method to explore the topologic efficiency abnormalities of WM structural connectome in asymptomatic adults with leukoaraiosis in the early stage. The main finding of our study was that WM injuries may demonstrate decreased kurtosis values in the ALAs. Second, in the ALA participants, global and local efficiency differences were found mostly in the fronto-limbic system regions compared with the healthy controls. Therefore, this metric could potentially be used as a quantitative parameter in the DKI technique to assess neuroimaging changes in the early stage of leukoaraiosis.

No differences in TBSS parameters were revealed in the asymptomatic subjects with leukoaraiosis. This may have been a result of the very small effect size of white matter changes and the limitation of sample size in our study.

The lower global and local efficiency of the brain network of participants in the ALA group in our study was similar to that of previously reported LA patients, which indicated that functional and structural integration in the LA stage was impaired [8,24]. The Eloc represented the fault tolerance of the network and Cp characterized the segregation ability of the network. The impaired efficiency of the brain network of the ALAs in our study, as displayed by a decreased Cp and decreased Eloc, indicated that the alterations may have been due to early WM degeneration; this finding is consistent with the previous studies that showed widespread WM degeneration in the early LA stage [25].

Specifically, we found that the nodal efficiency was impaired mostly disrupted in the left inferior frontal gyrus, left parahippocampal gyrus, left middle temporal gyrus, right calcarine fissure and surrounding cortex, and right temporal pole of the superior temporal gyrus. The inferior frontal gyrus, which participates in various creativity-relevant and memory processes, was reported to have a disrupted functional deficit as a result of neoplastic or neuropsychiatric disease [26,27]. The parahippocampal gyrus, which plays a role in high cognitive functions including memory encoding and visuospatial processes, was reported to have an effect on neurodegenerations in previous studies regarding leukoaraiosis [28]. The calcarine fissure includes most of the primary visual cortex and transfers the visual information through the ventral stream, which makes the CAL an influential region in healthy people [29,30]. Thus, the altered nodal efficiency of the CAL in our study suggested that visual impairment might occur in leukoaraiosis. However, this remains to be further experimentally demonstrated. Moreover, the temporal lobe was described in the early stages of neurodegenerative disorders. Temporal atrophy is associated with impaired cognitive function and neuropsychiatric symptoms in Alzheimer’s disease [31,32]. Recent studies suggested that Alzheimer’s disease degenerative pathologies contribute to white matter hyperintensity, in addition to cerebrovascular disease [33,34]. Therefore, the temporal lobe may be involved in neurodegeneration.

Hubs play a key role in the delivery of global information across the brain network, which is important to maintain the efficiency and stability of the network. However, they seem to be vulnerable and preferentially affected in patients with LA. Similar to previous findings, participants in both groups showed the most hubs in default mode, whereas some hub regions, such as the right insula and left middle temporal gyrus, diminished in the ALA group [24,35]. The reason for this may be due to the connection between hubs disrupted in the default network [24]. However, the balance between pathologic changes and connectivity or between these alterations and connectome disruption in the early stage of LA warrants further investigation.

Our study had several limitations. First, the sample size was relatively small as it was a preliminary study, which could reduce the statistical power of our analysis and lead to false-negative findings. Evaluation of a larger cohort is required. Second, we only explored white matter alterations, while grey matter structural networks could also be impacted in asymptomatic patients. A more comprehensive assessment of structural networks is needed to elucidate the pathogenesis of LA. Finally, deterministic tractography may cause fiber loss because of its intrinsic property. Future studies are needed to solve this problem with a probabilistic tractography approach.

## 5. Conclusions

We found some divergent topological network features using the graph analysis method in asymptomatic individuals with leukoaraiosis. This finding supported the idea that the DKI method may potentially be a clinically applicable measure by providing topologic characteristic changes for the early diagnosis of LA.

## Figures and Tables

**Figure 1 brainsci-12-00784-f001:**
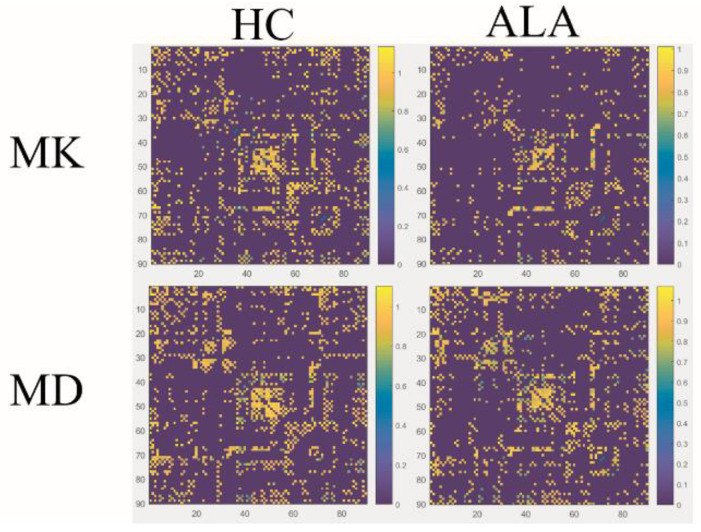
The interregional correlation matrix of mean diffusivity (MD) and mean kurtosis (MK) in the HC and ALA groups. The color bar indicates the value of the interregional parameter correlation. As is shown in the color maps, different dispersion in the MK metric were visually observed in the ALAs, but this was not observed in the MD metric.

**Figure 2 brainsci-12-00784-f002:**
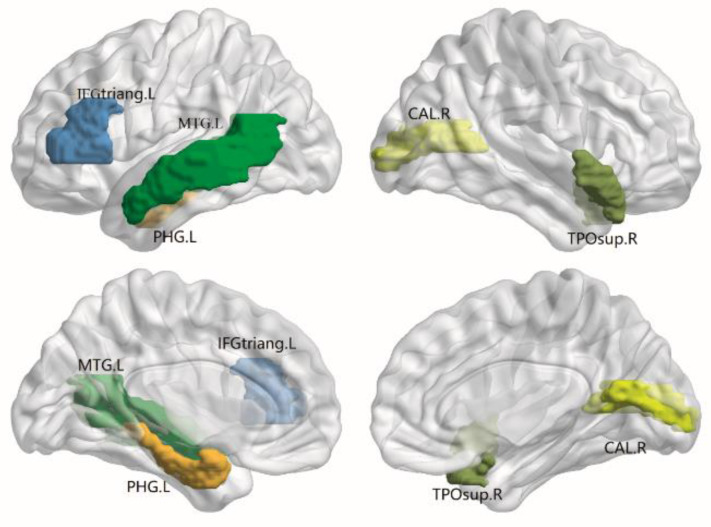
The distribution of brain regions with significant differences in nodal efficiency in FN-weighted white matter networks between the ALA and HC groups. IFGtriang.L: left triangular part of inferior frontal gyrus; MTG.L: left middle temporal gyrus; PHG.L: left parahippocampal gyrus; CAL.R: right calcarine fissure and surrounding cortex; TPOsup.R: right temporal pole of superior temporal gyrus.

**Figure 3 brainsci-12-00784-f003:**
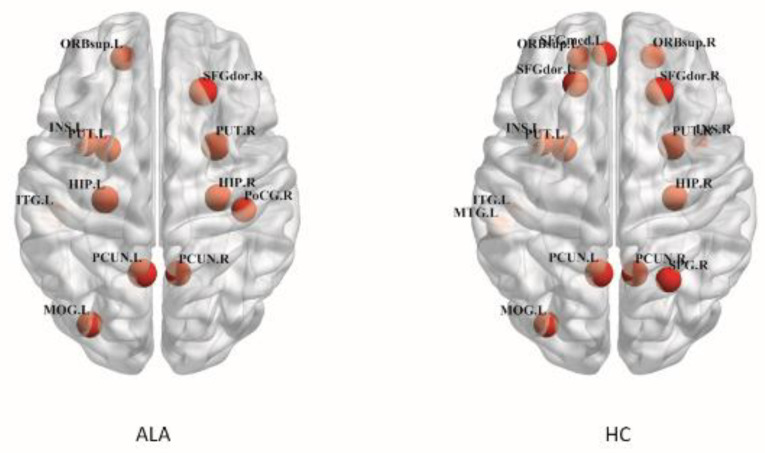
The distribution of hub regions in participants in the HC and ALA groups. HC: healthy control; ALA: asymptomatic subject with leukoaraiosis. The figure was processed using BrainNet Viewer software. “R” and “L” indicate the right and left sides, respectively. SFGdor: superior frontal gyrus, dorsolateral; SFGmed: superior frontal gyrus, medial; ORB.sup: superior frontal gyrus, medial orbital; INS: insula; PUT: putamen; HIP: hippocampus; ITG: inferior temporal gyrus; PoCG: postcentral gyrus; PCUN: precuneus; MOG: middle occipital gyrus; MTG: middle temporal gyrus; SPG: superior parietal gyrus.

**Table 1 brainsci-12-00784-t001:** Demographic data of subjects.

	ALA (*n* = 35)	HC (*n* = 20)	*p*-Value
Age	63 (8.86)	59 (9.22)	0.353
Education years	10.11 (2.44)	9.55 (1.97)	0.308
Sex	21 (60%)	10 (56%)	0.777
Hypertension	10 (38.5%)	6 (33.3%)	0.911
MMSE	28.45 (0.70)	28.79 (0.79)	0.117
MoCA	28.28 (0.85)	28.73 (0.80)	0.145

MMSE: Mini-Mental State Examination; MoCA: Montreal Cognitive Assessment. Values are means (standard deviation) for continuous variables, except for numbers for sex (% men) and vascular comorbidity (% yes).

**Table 2 brainsci-12-00784-t002:** Altered small-world characters of DKI metrics of the white matter network between the ALA and control groups.

		KT				DT			
		MK	KFA	AK	RK	MD	FA	AD	RD
Lp	ALA	4.11 (0.5)	8.99 (1.31)	6.71 (1.1)	4.95 (0.74)	3.81 (0.51)	11.98 (3.11)	3.77 (0.42)	4.61 (0.64)
	HC	4.26 (0.62)	9.03 (1.12)	6.86 (1.03)	4.79 (0.92)	3.96 (0.62)	12.04 (3.52)	3.86 (0.52)	4.52 (0.71)
Cp	ALA	0.23 (0.01) *	0.17 (0.02)	0.18 (0.10) *	0.13 (0.03) *	0.12 (0.01)	0.17 (0.01)	0.13 (0.13)	0.13 (0.01)
	HC	0.28 (0.01)	0.18 (0.02)	0.22 (0.13)	0.18 (0.02)	0.13 (0.01)	0.18 (0.01)	0.13 (0.12)	0.13 (0.01)
Eg	ALA	0.2 (0.12)	0.1 (0.07)	0.14 (0.02)	0.21 (0.13)	0.24 (0.02)	0.13 (0.02)	0.31 (0.18)	0.21 (0.06)
	HC	0.22 (0.01)	0.1 (0.02)	0.15 (0.01)	0.21 (0.02)	0.25 (0.00)	0.13 (0.01)	0.32 (0.01)	0.23 (0.35)
Eloc	ALA	0.41 (0.32) *	0.13 (0.02) *	0.31 (0.21) *	0.48 (0.03) *	0.50 (0.02)	0.17 (0.02)	0.56 (0.02)	0.44 (0.26)
	HC	0.44 (0.01)	0.17 (0.01)	0.34 (0.15)	0.53 (0.02)	0.51 (0.01)	0.17 (0.01)	0.57 (0.01)	0.44 (0.13)
σ	ALA	1.50 (0.11)	1.44 (0.13)	1.78 (0.01)	1.45 (0.31)	1.49 (0.21)	1.31 (0.11)	1.29 (0.14)	1.64 (0.34)
	HC	1.47 (0.13)	1.43 (0.13)	1.76 (0.04)	1.33 (0.15)	1.57 (0.17)	1.30 (0.11)	1.31 (0.13)	1.72 (0.27)

* means *p* < 0.05, corrected. Lp: characteristic path length; Cp: clustering coefficient; Eg: global efficiency; Eloc: local efficiency; σ: small-worldness. Scores are shown as mean (+SD).

## Data Availability

The datasets generated and/or analyzed during the current study are available from the corresponding author upon reasonable request.

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
