# Peer review of "Topologic Efficiency Abnormalities of the Connectome in Asymptomatic Patients with Leukoaraiosis"

_brainsci, 2022, doi:10.3390/brainsci12060784_

Round 1
Reviewer 1 Report
This manuscript explored the distruption of the brain network in asymptomatic patients with leukoaraiosis.
Some points should be modified for a better understanding of this publication.
Firstly, more has to be said about the patients recruitment. The only information which is given is that patients are enrolled from another study about neurodegenerative disorders. How were these asymptomatic patients recruited ? Did the authors scan a lot of subjects and selected only those with leukoaraisos ?
Secondly, the materials and methods part needs to be improved. Little information is given about what are Cp, Lp, Eg, Eloc or sigma used in table 2. Or about betweeness (line 177). Without a correct explanation of what is done, it is nearly impossible to evaluate the work done
Thirdly, line 152 deals with nodal metrics. Statistical threshold is set to p<0.005 uncorrected whereas in the whole document statistics are presented with correction (line 114). The overall conclusion of this paper may be different if you consider, as in line 158, that there is no result which reaches significance. My advise is to be as clear as possible in the Material and methods : it is impossible to define a statistical thresold in the methods and forget it if the results do no suit your hypotheses.
Some more minor points are to be corrected.
- Please check all the abreviations. For instance, on line 50, DKI is not defined.
- Please also check the bibliography (I noticed two typos : Lle Bihan instead of Le Bihan on ref 9, year not in bold in ref 8)
- Please also check for typos/english (Line 129 for instance)
- p-value for hypertension is missing in table 1, units are missing in table 1
Author Response
Dear reviewer,
thank you for your valuable comments.
All subjects were originally been recruited from community in another study concerning patients with neurodegenerative disease which we were also involved in. Yes, we picked out the eligible subjects with leukoaraisos as our asymptomatic group. Since that study has not been published so far, so we could not cite it.
We have expanded the materials and methods part with more information and references. And we also revised the Statistical analysis part and conclusion. And some minor mistakes have been corrected.
Reviewer 2 Report
In this manuscript, the authors measure diffusion-MRI and structural brain connectivity features and compare them between a small cohort of asymptomatic patients with leukoaraiosis and healthy controls, and present their findings. The study could be interesting for the neuroscientific community, but the methods need to be better described.
A clearer explanation is needed on how different networks were made for MK, AK, etc. Were tracts weighted by these measures?
I suggest that a subsection be allocated to the details of the performed correction for multiple comparisons in the statistical analyses, as this can highly affect the interpretation of the results.
There are quite a number of network features that could be studied. How were the specific features presented in this manuscript chosen?
If there is a reference for the other study from which the subjects were enrolled, it should be cited.
The word “brain” may be removed from the title, as “connectome” already implies it.
TBSS has been defined twice. Many other typos need to be fixed, e.g.: “an fractional anisotropy” --> a fractional anisotropy; “annd” --> and; “weighted of edge” --> weighting of edge; “to compared” --> to compare; “might occurs” --> might occur; “this still remain” --> this still remains
Author Response
Dear reviewer,
many thanks for your comments. They were all highly appreciated and helpful to us.
The title has been changed. Since the study from which the subjects were enrolled has not been published so far, so we could not cite it. We have expanded the materials and methods part with more information and references. In the revised part, we described the function or definition of the parameters we studied, e.g. Cp, Lp, Eg, Eloc or sigma. And some minor mistakes have been corrected.
Reviewer 3 Report
This paper applied DKI to explore the network difference between HC and patients with leukoaraiosis. While entire manuscript seems to be fluent, I still have several comments:
(1) Introduction mentioned few studies have focused on the early stage of LA. But it is unclear if this study was performed on the early stage of LA or not.
(2) Method section: the processing of DKI is too concise. Please provide details of the processing steps.
(3) Based on the description, I guess the network parameters were calculated based on the weighted network. Did you threshold the connectivity graph to remove some weak or random connections?
(4) Table 2: KT and DT is unclear to readers.
(5) For nodal-level results, the reported results are uncorrected. Though I understand the sample size might be the issue, you can still provide the raw p values of each results. Thus, we can determine if it is meaningful or not. If all the p values are just around 0.05, then the results might not be significant.
(6) Please be careful about the usage of "biomarker". It will require strict validation process and provide accuracy, specificity, sensitivity, et al.
Author Response
Dear reviewer,
many thanks for your comments. They were all highly appreciated and helpful to us.
We have add the missing Fazekas grade in the introduction to make clear that this study was performed on the early stage of LA. And we also have expanded the materials and methods part with more information and references. For nodal-level results, we provided the raw p values of each results. The conclusion was re-written to avoid the usage of "biomarker"